# One-Step Route to Fe_2_O_3_ and FeSe_2_ Nanoparticles Loaded on Carbon-Sheet for Lithium Storage

**DOI:** 10.3390/molecules27092875

**Published:** 2022-04-30

**Authors:** Denghu Wei, Leilei Xu, Zhiqi Wang, Xiaojie Jiang, Xiaxia Liu, Yuxue Ma, Jie Wang

**Affiliations:** School of Materials Science and Engineering, Liaocheng University, Liaocheng 252059, China; xllxf096364@163.com (L.X.); 13898872671@163.com (Z.W.); j15553292961@163.com (X.J.); 18853048100@163.com (X.L.); mayuxue1105@163.com (Y.M.)

**Keywords:** lithium-ion battery, anode materials, Fe_2_O_3_@C composite, FeSe_2_@C composite

## Abstract

Iron-based anode materials, such as Fe_2_O_3_ and FeSe_2_ have attracted widespread attention for lithium-ion batteries due to their high capacities. However, the capacity decays seriously because of poor conductivity and severe volume expansion. Designing nanostructures combined with carbon are effective means to improve cycling stability. In this work, ultra-small Fe_2_O_3_ nanoparticles loaded on a carbon framework were synthesized through a one-step thermal decomposition of the commercial C_15_H_21_FeO_6_ [Iron (III) acetylacetonate], which could be served as the source of Fe, O, and C. As an anode material, the Fe_2_O_3_@C anode delivers a specific capacity of 747.8 mAh g^−1^ after 200 cycles at 200 mA g^−1^ and 577.8 mAh g^−1^ after 365 cycles at 500 mA g^−1^. When selenium powder was introduced into the reaction system, the FeSe_2_ nano-rods encapsulated in the carbon shell were obtained, which also displayed a relatively good performance in lithium storage capacity (852 mAh g^−1^ after 150 cycles under the current density of 100 mA·g^−1^). This study may provide an alternative way to prepare other carbon-composited metal compounds, such as FeN_x_@C, FeP_x_@C, and FeS_x_@C, and found their applications in the field of electrochemistry.

## 1. Introduction

Over the last few decades, the ever-increasing demand for advanced energy storage devices has promoted the great success of lithium-ion batteries (LIBs), which have the advantages of high energy density and long cycle life and non-pollution, widely applied in electric vehicles and laptops [1,2,3,4]. However, the theoretical specific capacity of graphite as an anode material for lithium-ion batteries cannot meet future needs. Therefore, it is an urgent problem to develop high energy density and environmentally friendly anode materials [5].

The anode electrode materials are divided into the insertion-type (such as carbon, TiO_2_), the conversion-type (such as CoO, FeS_2_, NiSe_2_) as well as the alloy-type (such as Sn-based and Sb-based) according to the mechanism of the reaction [6]. Recently, the transition metal compounds as a class of conversion-type electrodes with outstanding theoretical specific capacity and high potential have attracted wide attention in the research direction of anode materials for lithium-ion batteries [7]. It is considered to be a new type of anode material with potential energy storage, and the research is gradually maturing.

Among all proposed electrode materials for LIBs, FeSe_2_ has important significance in the study due to the bandgap energy (E_g_ = 1.0 eV), and significant theoretical specific capacity (501 mAh g^−1^) [8,9,10], while FeSe_2_ has fewer related studies on lithium-ion batteries. Fe_2_O_3,_ as a transition metal oxide using a conversion mechanism to charge and discharge, compared with traditional graphite, is favored because of its high theoretical specific capacity(1004 mAh g^−1^), abundant resources, and environmental-friendly, which is considered to be a potential anode material for lithium ion batteries [11,12].

While similar to other transition metal oxides (TMOs) and transition metal sulfides (TMSs), FeSe_2_ and Fe_2_O_3_ had the limitations of low conductivity and serious volume expansion, resulting in the decrease of cycling stability. In order to ameliorate these shortcomings, carbonaceous materials, such as graphene, carbon nanotubes, and amorphous carbon shells, have been used to combine with FeSe_2_ and TMOs to improve the electrochemical performance [8,13,14]. For instance, Kong et al., reported that FeSe_2_@rGO could maintain over 900 mAh g^−1^ after 100 cycles at 0.1 A g^−1^ while only less than 40 mAh g^−1^ for bare FeSe_2_ anode in LIBs [8]. The incorporated carbon supports not only can act as a structural buffer to effectively alleviate the volume expansion but also significantly improve the conductivity.

Although there have been many reports on carbon composite metal compounds, there are still few simple methods to choose from. In this work, we reported a one-pot thermal decomposition of commercial Iron(Ⅲ) acetylacetonate (C_15_H_21_FeO_6_) to prepare Fe_2_O_3_@C composite (composed of Fe_2_O_3_ nanoparticles with a small size of around 50 nm loaded in and on the carbon supporter) for lithium storage (Figure 1). During the thermal decomposition process, the C_15_H_21_FeO_6_ can be served as the source for Fe, O, and C. When used as an anode material for LIBs, the Fe_2_O_3_@C composite could deliver a discharge capacity of 747.8 mAh g^−1^ after 200 cycles under the current density of 200 mA g^−1^, indicating relatively high stability during the charge-discharge processes. When selenium powder was introduced into the reaction system, the FeSe_2_ nano-rods encapsulated in the carbon shell were obtained, which also displayed a relatively good performance in lithium storage capacity (852 mAh g^−1^ after 150 cycles under the current density of 100 mA·g^−1^). This study suggests that the Fe_2_O_3_@C and FeSe_2_@C might be found for potential applications in LIBs, meanwhile, it provides an alternative way to prepare other carbon-composited metal compounds, and further work (such as FeN_x_@C, FeP_x_@C, and FeS_x_@C) is going on.

## 2. Experimental Section

### 2.1. Chemicals

All chemicals were analytical grade and without further purification. Iron(Ⅲ) acetylacetonate (C_15_H_21_FeO_6_) is purchased from Shanghai Macklin Biochemical Co., Ltd. (Shanghai, China); Selenium powder and NaCl are purchased from Sinopharm Chemical Reagent Co., Ltd. (Shanghai, China).

### 2.2. Synthesis of the Fe_2_O_3_@C Composite (Developed by Us)

Two grams of ferric acetylacetonate (C_15_H_21_FeO_6_) and 4 g of sodium chloride (NaCl) were dissolved in 15 mL absolute ethanol in a ball mill tank, adding two zirconia balls of varying sizes. After 8 h of ball milling, the uniform slurry was stirred in an oven at 80 °C for 12 h, then the mixture was pressed into a column and then heated to 600 °C for 5 h with the heating rate of 5 °C min^−1^ under N_2_ atmosphere. When the sample cooled down to room temperature, the excess salt was washed with water and ethanol. The Fe_2_O_3_@C was collected by dried 60 °C for 12 h.

### 2.3. Synthesis of the FeSe_2_ @C Composite (Developed by Us)

Two mmol of ferric acetylacetonate (C_15_H_21_FeO_6_), 4.4 mmol of Selenium powder (Se) and 1 g of sodium chloride (NaCl) were dissolved in 15 mL absolute ethanol to form uniform slurry in the same way. The FeSe_2_@C composite was obtained by annealing in the muffle furnace at 280 °C for 30 h with a heating rate of 2 °C min^−1^. The synthetic path of the samples is shown in Figure 1.

### 2.4. Materials Characterization

X-ray powder diffraction (XRD) patterns of the samples were recorded on a Philips X’ Pert Super diffract meter with Cu Kα radiation (*λ* = 1.54178 Å). The morphology of the product was observed on scanning electron microscopy (SEM, Merlin Compact, Carl Zeiss AG, Oberkochen, Germany). The elemental distribution of the sample was detected by energy-dispersive spectrometry (EDS) elemental mapping analysis (Merlin Compact). X-ray photoelectron spectroscopy (XPS) spectra were acquired on an ESCALAB 250 spectrometer. Thermogravimetric (TG) analysis was measured under air from room temperature to 700 °C with a heating rate of 10 °C min^−1^ by a TGA-2050.

### 2.5. Electrochemical Measurement

All relevant electrochemical tests were analyzed via CR2025 coin-type half cells in a glovebox filled with Argon atmosphere (H_2_O < 0.01 ppm, O_2_ < 0.01 ppm).

The working electrode was mixed with polyvinylidene-fluoride (CMC) and acetylene black at a weight ratio of 8:1:1 by ball-milling to form a homogeneous slurry with copper as a current collector and then dried in a vacuum at 60 °C for 12 h. The loading of the active material placed into each coin cell was approximately 1.0 mg. All cells were assembled by using the lithium disk as a counter electrode, the Celegard 2300 film as a separator and 1.0 M LiPF_6_ in ethylene carbonate (EC), ethyl methyl carbonate (EMC), and diethyl carbonate (DEC) as the electrolyte (EC/EMC/DEC = 1:1:1, *v*/*v*/*v*). The galvanostatic discharge-charge measurements between 0.01 and 3.0 V were conducted using a LAND-CT2001A battery tester. Cyclic voltammetry (CV) scanned from 0.01 to 3.0 V using an Electrochemical Workstation (CHI660E) was recorded at the scan rate of 0.1 mV s^−1^.

## 3. Results and Discussions

### 3.1. Composition and Microstructures of the Samples

The phase purities of the as-prepared samples were examined using X-ray powder diffraction (XRD), and the corresponding results are shown in Figure 2. The diffraction curve above is a FeSe_2_@C composite, all diffraction peaks correspond well to the characteristic peaks of the orthogonal phase FeSe_2_ (JCPDs No. 65-5270). Due to the strong diffraction peaks of FeSe_2_, the characteristic peaks (~26°) of carbon could not be observed [15]. In addition, there are no other superfluous peaks, indicating that the synthesized FeSe_2_@C composite is relatively pure. The diffraction line below is Fe_2_O_3_@C composite. The diffraction peaks centering at 30.2°, 35.6°, 43.2°, 57.2°, 62.9° could be designated to the (220), (311), (400), (511), (440) crystal planes of cubic phase Fe_2_O_3_ (JCPDs No. 39-1346). The diffraction peaks of carbon were not detected in the XRD, which might result from the low crystallinity of carbon [16]. No other diffraction peak proved the high purity of the sample.

Figure 3 shows the morphology and microstructure of the Fe_2_O_3_@C by SEM. From the picture, it is obvious that many Fe_2_O_3_ particles are uniformly embedded into the carbon skeleton. To further testify the distribution of elements, we characterized the energy dispersive EDS mapping in Figure 3b–e and attested that iron, oxygen and carbon elements are uniformly dispersed in the composite.

Figure 4a exhibits the SEM image of the FeSe_2_@C composite. It can be seen from the picture that the product is FeSe_2_ with an uneven rod-like structure coated by carbon. Moreover, the rod-like structure of the FeSe_2_ encapsulated by carbon was further observed through TEM images (Figure 6e,f).

The EDS element mapping results in Figure 4b–e indicated that iron, carbon and selenium were scattered equably in the composite. Energy dispersive spectroscopy (EDS) in Figure 4f revealed that the mass ratio of iron to selenium is about 1:2, further confirming the successful synthesis of FeSe_2_. In addition, the presence of the O element in the surface map conjectured that FeSe_2_ was oxidized at room temperature with the presence of trace Fe_x_O_y_.

To find out the exact carbon contents of the FeSe_2_@C and Fe_2_O_3_@C composites, thermogravimetry (TG) was employed from ambient temperature to 750 °C in an air atmosphere with a rate of 10 °C min^−1^ and the corresponding results are shown in Figure 5.

Figure 5a presents the TG curves of the FeSe_2_@C. In the range from 200 °C to 280 °C, the weight obviously increased due to the formation of SeO_2_ and Fe_2_O_3_ by reaction of the FeSe_2_ with oxygen in the air atmosphere. As the temperature gradually rises, it has a very rapid decline, which should be ascribed to the combustion of carbon and gasification of SeO_2_ in the air [15]. Based on the reaction formula: 4FeSe_2_ (s) + 11O_2_ (g) → 8SeO_2_ (g) + 2Fe_2_O_3_ (s). For the final Fe_2_O_3_ content of 27.6%, the content of FeSe_2_ loadings in the FeSe_2_@C composite was 73.8%. Therefore, the carbon content was 26.2%.

Figure 5b displays the thermogravimetric analysis of the Fe_2_O_3_@C composite. When the temperature increases from 300 °C to 450 °C, the mass lost was 31.8%, which can be attributed to the conversion of carbon in the air. The carbon content for the Fe_2_O_3_@C was calculated to be 31.8%. To determine the average size of Fe_2_O_3_ particles, TEM images were taken. As shown in Figure 6a–c, Fe_2_O_3_ particles with an average of around 50 nm were uniformly distributed in the amorphous carbon framework. Figure 6d–f presents the TEM images of the FeSe_2_@C sample. The thickness of the carbon coating on FeSe_2_ rods along the edges was about 10 nm. As the characteristic peak of carbon is not presented in both XRD of the two samples, Raman spectra (Figure 6g,h) were taken to further indicate that the carbon is amorphous in nature.

The elemental compositions and valence states of the FeSe_2_@C and Fe_2_O_3_@C composites were examined by X-ray photoelectron spectroscopy (XPS) as shown in Figure 7. According to the overall survey spectrum, the existence of the Fe, Se, C and O elements was confirmed in Figure 7a. A couple of peaks at 718.8 eV and 706.3 eV (Figure 7b), are associated with Fe 2p_1__/2_ and Fe 2p_3__/2_, implying the existence of Fe^2+^ [17,18]. In addition, the Fe 2p spectra show a peak at 710.5 eV due to exposure to air to form a small amount of Fe_2_O_3_, which corresponds to the results of previous reports for FeSe_2_ [19]. Two characteristic peaks located at 54.2 eV and 53.6 eV in Figure 7c, corresponding to Se 3d_5/2_ and Se 3d_3/2_ [20], further proved that the expected product was FeSe_2_. Based on the results of XPS, we can know that the FeSe_2_@C composite has been successfully synthesized.

The survey spectrum of the Fe_2_O_3_@C composite (Figure 7d) revealed that it was composed of Fe, O and C elements. The peaks at 724.2 eV and 710.2 eV depicted in Figure 7e could be assigned to Fe 2p_1/2_ and Fe 2p_3/2_, respectively, which was in agreement with the previous reports for Fe_2_O_3_.

### 3.2. Electrochemical Property in Half-Cells

To further study the electrochemical properties of the obtained composites, the samples were assembled into button cells and characterized by a series of electrochemical tests. All tested specific capacity is calculated according to the Fe_2_O_3_@C and the FeSe_2_@C load. The electrochemical test results are shown in Figure 8 and Figure 9.

The first three cyclic-voltammetry (CV) curves of the Fe_2_O_3_@C were measured in the voltage window from 0.01 to 3.0 V with the scan rate of 0.1 mV·s^−1^(Figure 8a). In the first cathodic scan, the reduction peak located at 0.5 V can be ascribed to the formation of the SEI (solid electrolyte interphase) layer and the reduction from Fe_2_O_3_ to Fe^0^. The oxidation peak around 1.6 V was related to the oxidation of Fe^0^. The first curve is obviously different from the following two curves. The intensity of the redox peak is weakened, indicating that the capacity is decreasing, which was attributed to the irreversible capacity loss in the electrolyte. In addition, from the second cycle, the reduction peak moved back to a higher potential, while the position of the oxidation peak was basically unchanged, revealing that the process of Fe^0^ to Fe^3^^+^ is reversible. It is worth noting that the CV curves remained consistent for the second and third cycles, displaying the good electrochemical reversibility of the sample.

Figure 8b displays the 1st, 2nd and 3rd charge and discharge profiles of the Fe_2_O_3_@C composite at a current density of 200 mA g^−1^ with a voltage ranging from 0.01–3.0 V. We can observe that the first discharge curve appears as a platform around 0.5–0.8 V, corresponding to the reduction of Fe^3+^ to Fe^0^. Moreover the discharge platform moved to about 0.98 V in the subsequent discharge curve, which is attributed to irreversible reactions in the first cycle. For the initial charge curve, a sloping platform appears in 1.50 V–2.20 V related to the oxidation of Fe^0^ to Fe^3+^, consistent with literature reports [21]. In comparison with CV curves, we can note that these platforms correspond to the location of redox peaks in the CV curves. In addition, we can also intuitively see from Figure 8b that the first charge-discharge specific capacity of the Fe_2_O_3_@C is 837.6 mAh g^−1^ and 1227.8 mAh g^−1^, respectively. Moreover, the specific capacity of the second discharge dropped sharply to 809.9 mAh g^−1^. The loss of capacity may be assigned to the formation of the SEI film and further loss of lithium. What is more, the 2nd and 3rd charge and discharge curves are observed to be overlapping, showing the good electrochemical reversibility of the sample.

The cycling performance of this carbon-coated Fe_2_O_3_ composite was conducted at the current density of 200 mA g^−1^ (Figure 8c) and 500 mA/g (Figure 8d) within the voltage window of 0.01–3.0V. At a current density of 200 mA g^−1^, its initial charge and discharge capacities are 837.6 mAh g^−1^ and 1227.8 mAh g^−1^, respectively. So, its first Coulomb efficiency was 68.2%, mainly caused by the production of SEI film. After 200 cycles, discharge specific capacity can be up to 747.8 mAh g^−1^. At the current density of 500 mA g^−1^, the first charge-discharge specific capacity is 674.7 mAh g^−1^ and 1018.3 mAh g^−1^, respectively. After 365 cycles, the capacity still keeps at 577.8 mAh g^−1^, indicating good cycle stability. Due to carbon coating, the volume expansion of Fe_2_O_3_ nanoparticles may be effectively alleviated during the deintercalation of lithium, and carbon can improve the conductivity of the Fe_2_O_3_@C composite.

To study the lithium-ion storage behavior of the FeSe_2_ electrode, cyclic voltammetry (CV) of the first three cycles was firstly recorded at a scan rate of 0.1 mV s^−1^ in the voltage range of 0.01–3.0 V as shown in Figure 9a. From the diagram, there are two pairs of redox peaks, indicating the occurrence of two electrochemical reactions. In the first cathodic process, two reduction peaks around 1.13 V and 0.7 V were attributed to the generation of Li_x_FeSe_2_, the formation of the SEI layer decomposed by the electrolyte and the formation process of FeSe and Li_2_Se [22], respectively. Meanwhile, two oxidation peaks at 2.0 V and 2.2 V appeared in the initial anodic cycle, corresponding to the formation of Li_x_FeSe_2_ and FeSe_2_, severally. In addition, the cyclic voltammetry curves of the first lap are obviously different from the subsequent cycles, in which the formation of SEI film on the surface of the FeSe_2_@C electrode led to a lower Coulomb efficiency during the first charge-discharge process. In the subsequent cycles, the sharp peak at approximately 1.2 V shifts to 1.6 V, indicating the activation process in the first cycle. The redox peaks of the second and third cycles did not change significantly, in which lithium intercalation is only a fraction of the subsequent intercalation and removal of lithium ions, not affecting the internal structure of the FeSe_2_ during the charge and discharge process.

Figure 9b shows the charge/discharge curves of the FeSe_2_@C at a current density of 100 mA g^−1^ with a voltage range of 0.01–3.0 V. As we can see, the FeSe_2_@C composite has several obvious charging and discharging platforms, illustrating that there have the reversible reactions. There are reversible electrochemical reaction processes [8,13,22,23]:FeSe_2_ + x Li^+^ + x e^−^→ Li_x_FeSe_2_
Li_x_FeSe_2_ + x Li^+^ + x e^−^ →Li_2_Se + FeSe
FeSe + 2 Li^+^ + 2 e^−^ → Fe + Li_2_Se
Fe + Li_2_Se → Li_x_FeSe_2_ + (4 − x) Li^+^ + (4 − x) e^−^
Li_x_FeSe_2_ + x Li^+^ + x e^−^ → FeSe_2_ + x Li^+^ + x e^−^

The discharge voltage platforms for the first cycle of FeSe_2_@C nanomaterials are 0.8 V and 1.50 V and the charging voltage platform are 1.8 V and 2.2 V. The first charge/discharge specific capacity of the electrode was 547.9 mAh g^−1^ and 710.2 mAh g^−1^, respectively. We can observe that there is no obvious deviation between the first charging platform and the subsequent two cycles of the charging platform, but the first discharge curves are obviously different from the subsequent curves. The discharge capacity of the second and the third cycles was reduced to 574 mAh g^−1^, ascribed to irreversible processes, including the formation of SEI film on the surface of the electrode and the insufficient decomposition of electrolyte and Li_2_Se. Meanwhile, the discharge and charge curves for the FeSe_2_@C electrode remain stable and overlap very well from the second cycle, indicating that the reaction remains reversible and steady after the first cycle, consistent with the results of CV.

To explore the cycling stability of the FeSe_2_@C electrode, galvanostatic charge/discharge testing was employed at a current of 100 mA g^−1^ (Figure 9c). The first charge/discharge specific capacity is 547.9 mAh g^−1^ and 710.2 mAh g^−1^, respectively. After 150 cycles, discharge specific capacity up to 852mAh g^−1^. The increase of specific capacity may be due to the coating of carbon, which leads to the improvement of conductivity, or the incomplete reactions brought about by the presence of metal Se during selenylation.

## 4. Conclusions

In this report, the Fe_2_O_3_@C composite and FeSe_2_@C composite have been synthesized via a one-pot thermal decomposition of commercial salt. C_15_H_21_FeO_6_ served as the source for Fe, C and O. The content of carbon is 27.6% in Fe_2_O_3_@C, while the content of carbon is 31.8% in FeSe_2_@C. As the anode material for lithium-ion batteries, the Fe_2_O_3_@C electrode exhibited a reversible capacity of 747.8 mAh g^−^^1^ at 200 mA g^−1^ after 200 cycles, even under high current, its reversible capacity could reach 577.8 mAh g^−1^ after 360 cycles. What is more, the FeSe_2_@C electrode displayed a good capacity of 852 mAh g^−1^ at 100 mA g^−1^ after 150 cycles. The good cycling stability may be ascribed to the conductive carbon framework, which not only can promote the electrical conductivity, but relieve volume expansion during the charge/discharge process. This research suggests that the Fe_2_O_3_@C and FeSe_2_@C synthesized by the one-pot thermal decomposition route might be found for potential applications in LIBs, meanwhile, it provides an alternative way to prepare other carbon-composited metal compounds. Moreover, further work (such as FeN_x_@C, FeP_x_@C, and FeS_x_@C) is being carried out.

## Figures and Tables

**Figure 1 molecules-27-02875-f001:**
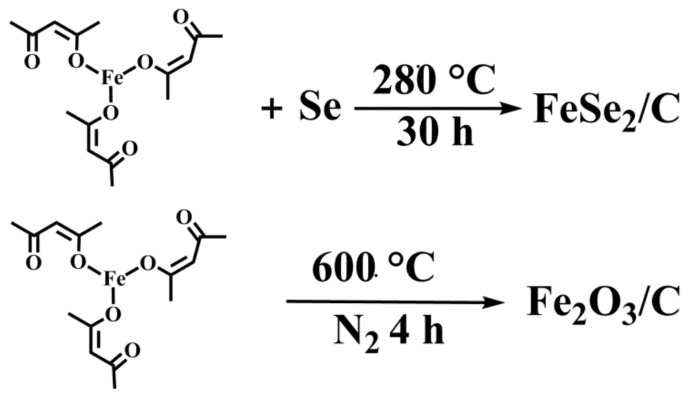
Schematic illustration for the preparation of FeSe2@C and Fe2O3@C composites.

**Figure 2 molecules-27-02875-f002:**
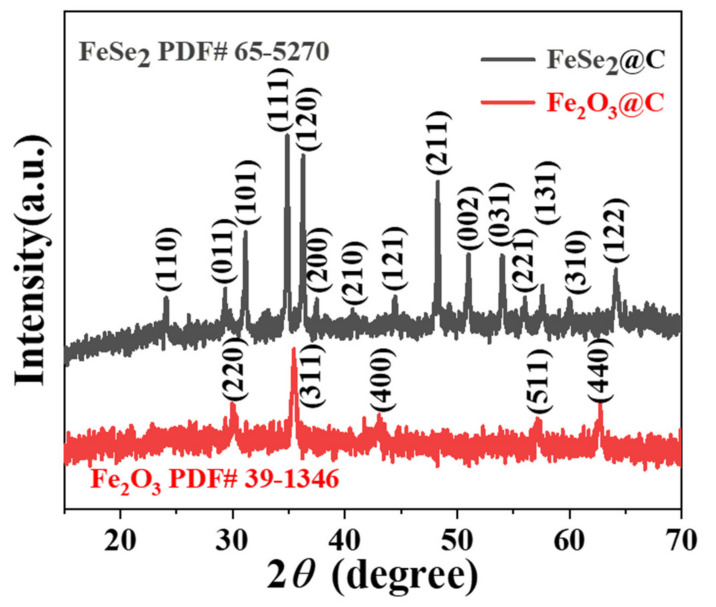
XRD patterns of FeSe_2_@C and Fe_2_O_3_@C composites.

**Figure 3 molecules-27-02875-f003:**
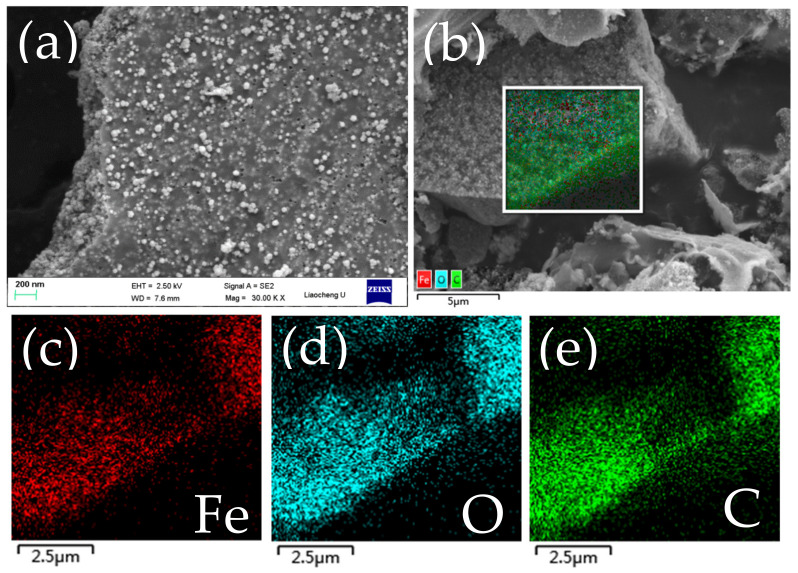
Fe_2_O_3_@C composite: (**a**) SEM image, (**b**) EDS hierarchical image, (**c**–**e**) EDS spectrum with elemental mapping.

**Figure 4 molecules-27-02875-f004:**
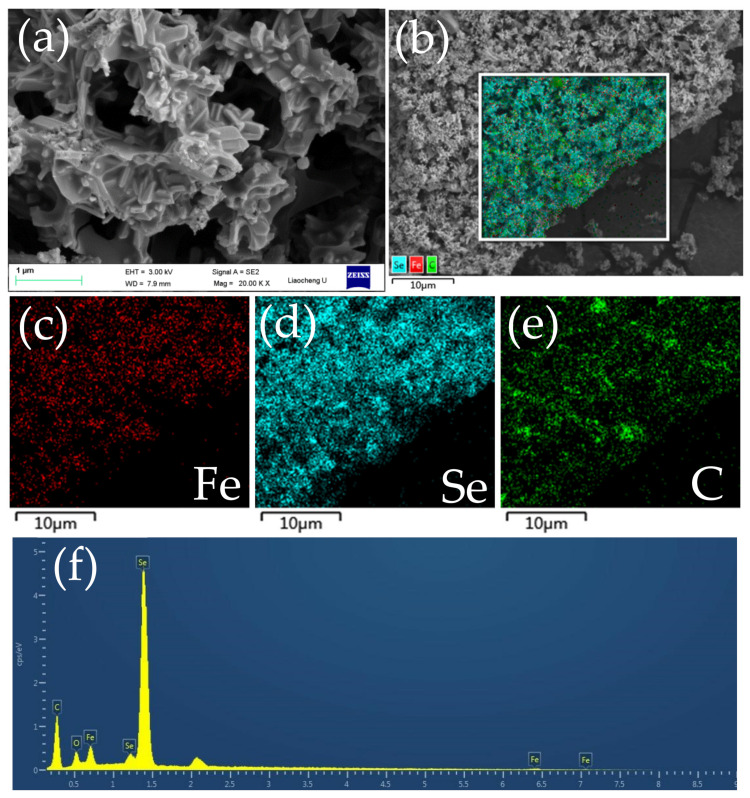
Fe_2_O_3_@C composite: (**a**) SEM image, (**b**) EDS hierarchical image, (**c**–**e**) EDS spectrum with elemental mapping; (**f**) total surface spectrum.

**Figure 5 molecules-27-02875-f005:**
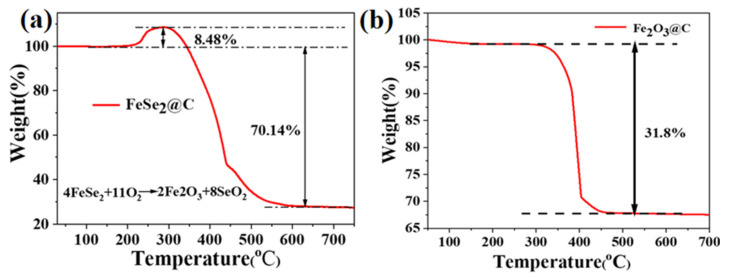
(**a**) TG curve of FeSe2@C composite; (**b**) TG of Fe2O3@C composite.

**Figure 6 molecules-27-02875-f006:**
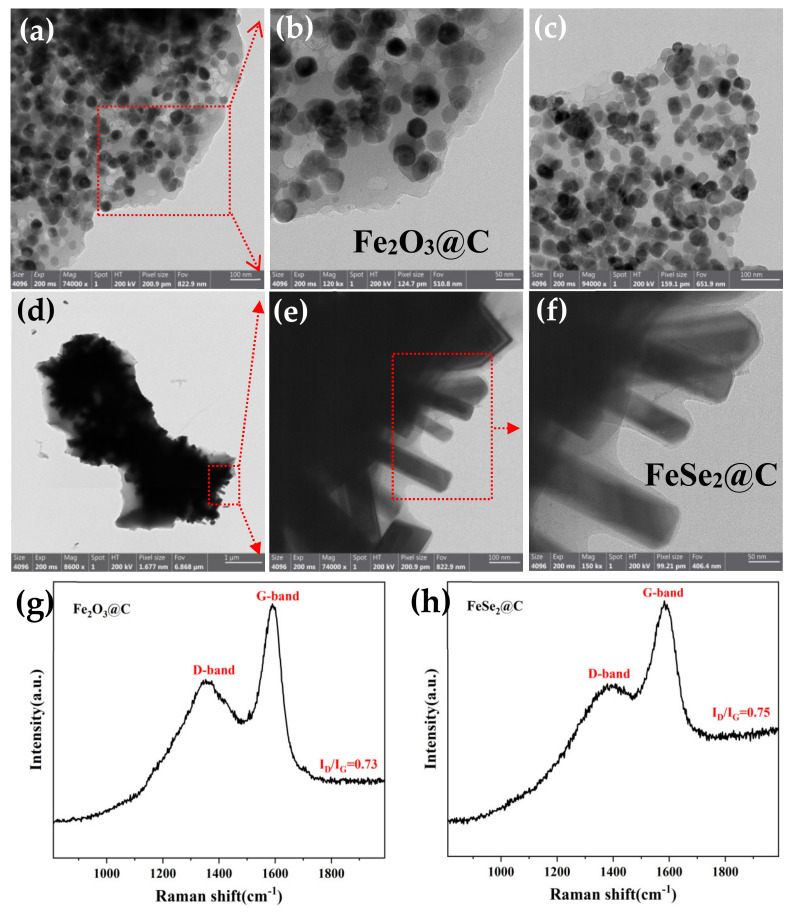
TEM images of Fe_2_O_3_@C (**a**–**c**) and FeSe_2_@C (**d**–**f**); Raman spectra of Fe_2_O_3_@C (**g**) and FeSe_2_@C (**h**).

**Figure 7 molecules-27-02875-f007:**
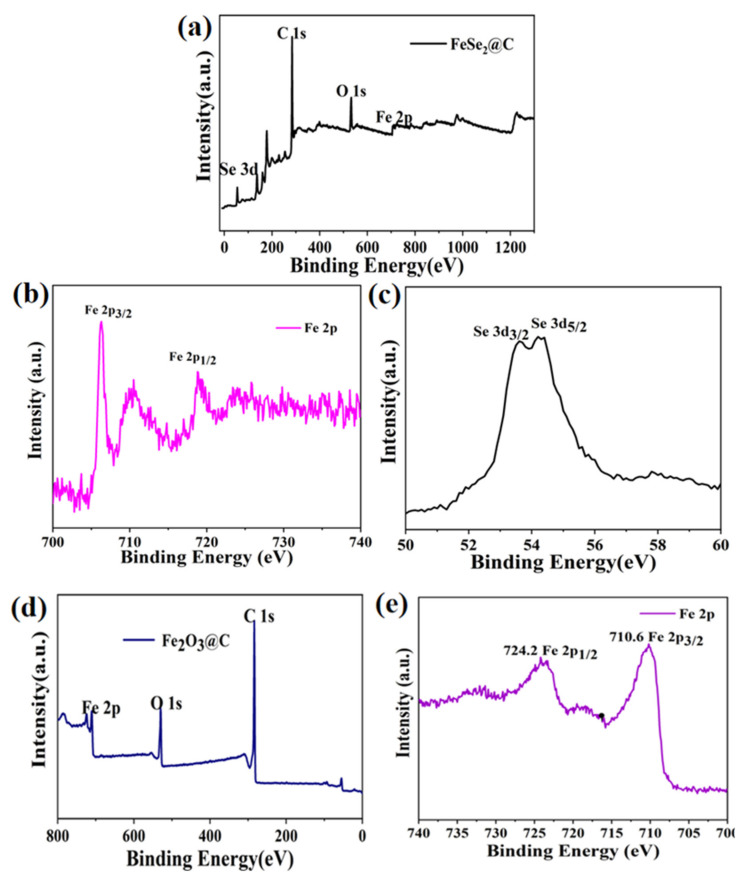
(**a**–**c**) XPS spectra of FeSe_2_@C composite: (**a**) survey spectrum, (**b**) Fe 2p region, (**c**) Se 3d region; (**d**,**e**) XPS spectra of Fe_2_O_3_@C composite (**d**) survey spectrum, (**e**) Fe 2p region.

**Figure 8 molecules-27-02875-f008:**
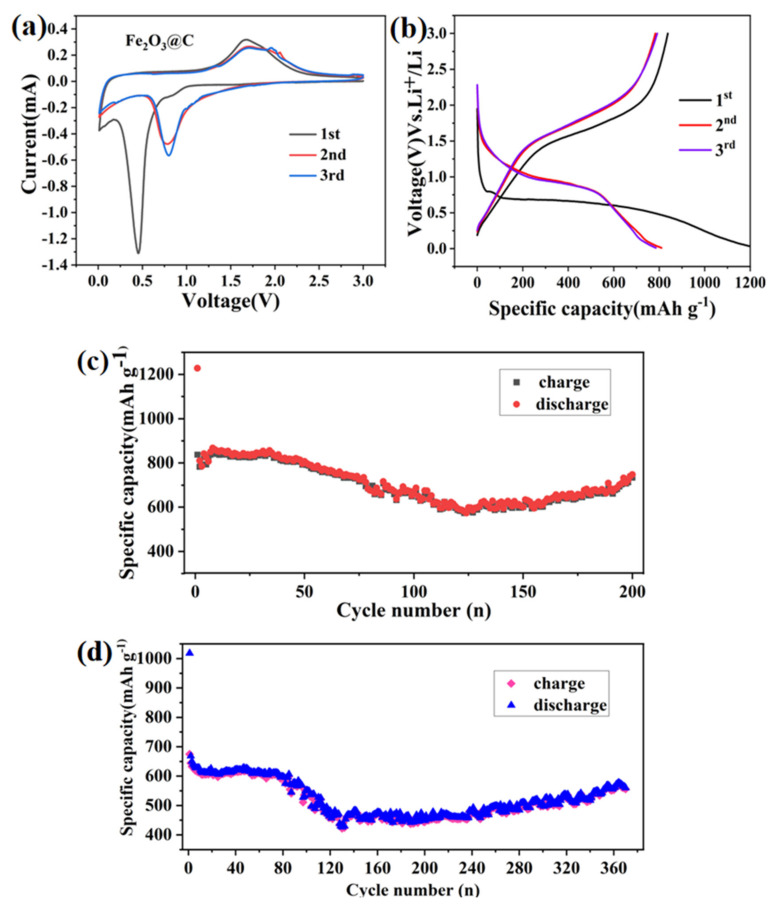
Electrochemical properties of the Fe_2_O_3_@C electrode: (**a**) Cyclic voltammograms between 0.01 and 3 V at a potential sweep rate of 0.1 mV s^−1^. (**b**) Discharge/charge profiles at 0.2 A g^−1^, and (**c**,**d**) Cycling performance at 0.2 A g^−1^ and 0.5 A g^−1^, respectively.

**Figure 9 molecules-27-02875-f009:**
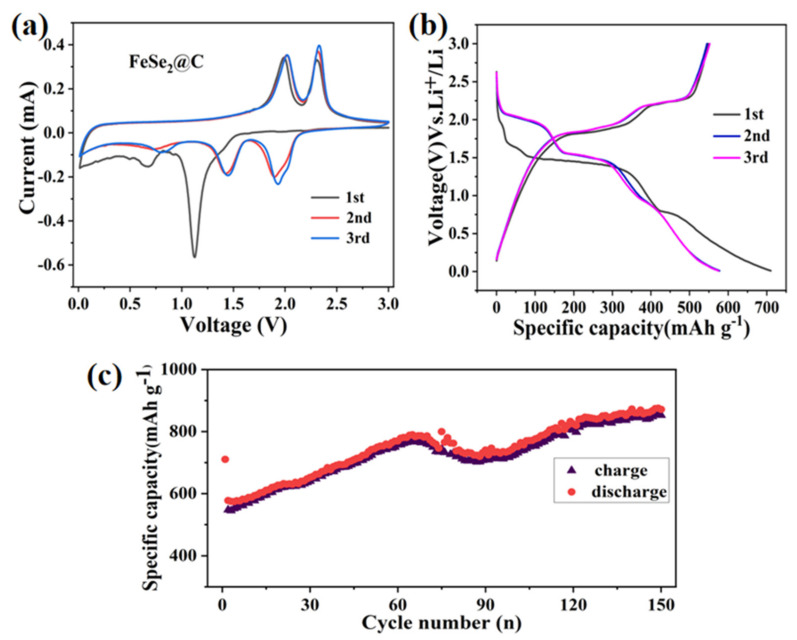
Electrochemical properties of the FeSe_2_@C electrode: (**a**) Cyclic voltammograms between 0.01 and 3 V at a potential sweep rate of 0.1 mV s^−1^. (**b**) Discharge/charge profiles at 0.1 A g^−1^, (**c**) Cycling performance at 0.1 A g^−1^.

## Data Availability

Not applicable.

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
