# Peer review of "One-Step Route to Fe2O3 and FeSe2 Nanoparticles Loaded on Carbon-Sheet for Lithium Storage"

_molecules, 2022, doi:10.3390/molecules27092875_

Round 1

Reviewer 1 Report

The present manuscript shows the application of  Fe2O3 and FeSe2 as an anode material. The prepared material was checked with instruments such as  X-ray differaction patterns,  SEM-EDS,  elemental mapping,  TEM, Raman, and  XPS. The characterization part is fine but the materials application parts need more information. In my opinion the submitted manuscript deserve publication after revision with more informations. below is some suggestions to improve their manuscript.    

  1. Abstract need conclusion part at the end of the abstract.  "What’s more, the synthetic strategy can be simply extended to prepare other iron-based anode materials, FeSe2@C for instance also show good cycling performance" only this sentence is not enough. 
  2. Authors need to check the Temperature-Programmed Oxidation (TPO) for their samples to analyse the surface reaction with metal and carbon.  
  3. Authors also need to analyse the Temperature-Programmed Reduction (TPR) because TPR is utilized to examine the reducibility, the metal–support interaction, and to find the activation and/or reduction temperature that results in metallic particles required for the catalytic reaction or surface reaction.     
  4. Authors need to add mechanism which show the relation between the materials bonding with carbon surface. There is no mechanism from which readers can understand well.  
  5. conclusion need to elaborate with the detail of the obtained results also provide the future directions of the present study. 

Reviewer 2 Report

The authors made a very interesting work. Due the fact of increasing need of various wearable appliances or gadgets, this topic is very important to whole population. Paper is well written, but some sentences could have been written shortly and the authors must short them. I found two small mistakes:

  • There is no information about used chemicals (producer, country of origin, etc.)
  • Synthesis processes are not clear. Did the authors developed them or found in literature? It was not mentioned.
  • Line 85: there is no any information about producer of TG, please add
  • Figures 3, 4 and 5 have wrong titles, I suppose it should be: title of Fig 3 belongs to Fig 4;  title of Fig 4 belongs to Fig 5 and title of Fig 5 belongs to Fig 3. The authors must rearrange and correct   

Reviewer 3 Report

Comments and Suggestions for Authors

Dear Authors,

The Title:

One-Step Route to Fe2O3 and FeSe2 Nanoparticles Loaded on Carbon-Sheet for Lithium Storage

I have to read your manuscript with great attention and interest.

As part of the research, iron compounds were synthesized for use in lithium-ion batteries.

The submission falls within the scope of the journal and is sufficiently original. I recommended the publication in a present form.

Present the research of other authors in more detail in order to be able to compare your results with other science centers

Has the material been corroded as has been checked for stability?

Suggest your mechanism of action of iron compounds

Delete in the numbering of equations 4-1: 1

Reviewer 4 Report

Thanks for the invitation to review the manuscript which can be published but require major revision

  1. Introduction authors should clearly mention why they prepare these kind of materials. I know there are many studies related with Fe2O3 and FeSe2 so what is making this work interetsing.
  2. Caption of Figure 3 is not correct. Authors need to recheck it. Infact they did not discuss about TGA as well.

  3. Authors mentioned that FeSe2@C has rod-like structure, however, SEM images does not confirm it.

  4. figure 5 does not have TGA data.  Check figure number carefully.

  5. In battery experiment, I could not see any controlled experiment which can be compared with Fe2O3 or FeSe2 alone and should show advantage of it carbon composite.
  6. Some important refernces should be included in manuscript Small, 2021, 17 (4), 2006651; Journal of Materials Science volume 54pages4225–4235 (2019); Chem. Commun., 2019,55, 10960-10963.

Round 2

Reviewer 4 Report

Authors revised manuscript well. Therefore, it can be published in its current form.